# Perturbing BatchNorm and Only BatchNorm Benefits Sharpness-Aware Minimization

**Maximilian Müller**
University of Tübingen and Tübingen AI Center
`maximilian.mueller@wsii.uni-tuebingen.de`

**Matthias Hein**
University of Tübingen and Tübingen AI Center
`matthias.hein@uni-tuebingen.de`

## Abstract

We investigate the connection between two popular methods commonly used in training deep neural networks: Sharpness-Aware Minimization (SAM) and Batch Normalization. We find that perturbing *only* the affine BatchNorm parameters in the adversarial step of SAM benefits the generalization performance, while excluding them can decrease the performance strongly. We confirm our results across several models and SAM-variants on CIFAR-10 and CIFAR-100 and show preliminary results for ImageNet. Our results provide a practical tweak for training deep networks, but also cast doubt on the commonly accepted explanation of SAM minimizing a sharpness quantity responsible for generalization.

## 1 Introduction

The generalization of deep neural networks has been linked to the flatness of the loss-surface already by [8]. In the past years, several studies attempted to better understand this relation [10, 5, 11, 16] and to leverage it for improved learning algorithms [19]. Most notably, [6] proposed Sharpness-Aware Minimization (SAM), an optimization technique designed towards directly minimizing a sharpness-based generalization bound alongside the training loss and achieved state-of-the-art in a variety of vision-benchmarks. SAM minimizes an adversarially perturbed loss, where the perturbation is computed with respect to a small $l_2$-ball around the current point in the optimization trajectory. Follow-up works tried to either reduce the computational cost of the method [4], or improve its performance. [20] minimize a different objective called *surrogate gap*, whereas others aim at defining a more accurate perturbation model, which is e.g. adaptive to the scale of the parameters and hence invariant to reparametrizations of the network (ASAM [14]) or respects the parameter space geometry induced by the Fisher information (Fisher-SAM [12]).

While the empirical success of SAM-like methods is commonly attributed to them finding favorable flat minima, there remain doubts on whether this explanation can provide a conclusive picture. [1] argue that the generalization bound, which provides the main theoretical justification for the SAM algorithm, is based on average-case sharpness, but SAM with random instead of worst-case perturbations does not substantially improve over SGD. Further, the empirical success of m-sharpness, a gradient averaging heuristic explained in Section 2, and the observation that a more accurate optimization of the perturbation model decreases SAMs generalization performance (discussed in [6]) additionally weakens this reasoning. [1] hypothesize that some other quantity, possibly correlated with (m-)sharpness, could be responsible for generalization and the main success of SAM might instead be due to its benefitial implicit bias.

Has it Trained Yet? Workshop at the Conference on Neural Information Processing Systems (NeurIPS 2022).

We therefore take a step back from the sharpness-explanation and instead investigate the interplay between SAM and Batch Normalization (BN) [9], another technique that has shown to be crucial for training well-generalizing networks, but remains poorly understood. [17] showed that the initial explanation of BN-layers mitigating a covariate shift in the activations is insufficient, and instead linked it to a smoothing of the loss-surface, whereas [15] understood it as an implicit regularization scheme. [7] showed that the affine parameters of the layer are by themselves already surprisingly expressive: Training only $\gamma$ and $\beta$ of deep ResNets and freezing all other parameters at their random initialization yielded non-trivial accuracy on CIFAR-10 and ImageNet, which could not be achieved when training an equivalent number of other parameters.

In this work, we show that ASAM, as it was proposed in [14], relies crucially on the BatchNorm parameters and that computing the adversarial perturbation without them degrades its performance strongly. We further demonstrate for a range of models and SAM-variants, that perturbing *only* the BatchNorm parameters in the adversarial step typically boosts their performance, leading to a simple adjustment that can easily be incorporated in existing SAM-like methods.

## 2    Background: Sharpness-Aware Minimization

We recapitulate SAM, ASAM and Fisher-SAM with their respective perturbation models. To this end, we consider a neural network $f_{\mathbf{w}} : \mathbb{R}^d \longrightarrow \mathbb{R}^k$ which is parameterized by a vector $\mathbf{w}$ as our model. The train dataset consists of pairs $S^{train} = \{(\mathbf{x}_1, \mathbf{y}_1), ...(\mathbf{x}_n, \mathbf{y}_n)\}$ which are drawn from the data distribution $D$ and we write the loss function as $l : \mathbb{R}^k \times \mathbb{R}^k \longrightarrow \mathbb{R}_+$. The goal is to learn a model $f_{\mathbf{w}}$ with good generalization performance, i.e. low expected loss $L_D(\mathbf{w}) = \mathbb{E}_{(\mathbf{x},\mathbf{y}) \sim D}[l(\mathbf{y}, f_{\mathbf{w}}(\mathbf{x}))]$ on the distribution $D$. The training loss can be written as $L(\mathbf{w}) = \frac{1}{n} \sum_{i=1}^{n} l(\mathbf{y}_i, f_{\mathbf{w}}(\mathbf{x}_i))$. Conventional SGD-like optimization methods minimize (a regularized version of) $L$ by stochastic gradient descent. SAM aims at additionally minimizing the worst-case sharpness of the training loss in a neighborhood defined by an $l_p$ ball around $\mathbf{w}$, i.e. $\max_{||\epsilon||_p < \rho} L(\mathbf{w} + \epsilon) - L(\mathbf{w})$. This leads to the overall objective

$$\min_{\mathbf{w}} \max_{||\epsilon||_p < \rho} L(\mathbf{w} + \epsilon). \tag{1}$$

In practice, SAM uses $p = 2$ and approximates the inner maximization by a single gradient step, yielding $\epsilon = \rho \nabla L(\mathbf{w}) / ||\nabla L(\mathbf{w})||_2$ and requiring an additional forward-backward pass compared to SGD. The gradient is then re-evaluated at the perturbed point $\mathbf{w} + \epsilon$, giving the actual weight update:

$$\mathbf{w} \longleftarrow \mathbf{w} - \alpha \nabla L(\mathbf{w} + \epsilon) \tag{2}$$

Computing $\epsilon$ separately for each GPU in multi-GPU settings and then averaging the resulting perturbed gradients for the update step in (2) has been shown to increase SAMs performance. This method is called m-sharpness, with m being the number of samples on each GPU. Since the perturbation model in (1) is not invariant to rescaling of $f_{\mathbf{w}}$ [3], ASAM [14], a partly scale-invariant version of SAM, was proposed, with the objective

$$\min_{\mathbf{w}} \max_{||T_w^{-1}\epsilon||_p < \rho} L(\mathbf{w} + \epsilon) \tag{3}$$

where $T_w$ is a normalization operator, making the perturbation adaptive to the scale of the network parameters. [14] choose $T_w$ to be diagonal with entries $T_w^i = |w_i| + \eta$ for weight parameters and $T_w^i = 1$ for bias parameters, and $\eta$ is typically set to $0.01$. Equivalently to SAM, the inner maximization is solved by a single gradient step:

$$\epsilon_2 = \rho \frac{T_w^2 \nabla L(\mathbf{w})}{||T_w \nabla L(\mathbf{w})||_2} \text{ for } p = 2, \qquad \epsilon_\infty = \rho T_w \text{sign}(\nabla L(\mathbf{w})) \text{ for } p = \infty \tag{4}$$

We note that for $T_w$ being the identity matrix, this is equivalent to the SAM formulation. Recently, [12] proposed to use a distance metric induced by the Fisher information instead of a Euclidean distance measure between parameters. The approach can also be framed as a variant of ASAM, with $T_w$ being diagonal with entries $T_w^i = 1/\sqrt{1 + \eta f_i}$ and $f_i$ approximating the $i^{th}$ diagonal entry of the Fisher-matrix by the squared average batch-gradient, $f_i = (\partial_{w_i} L_{Batch}(\mathbf{w}))^2$. For our experiments, we additionally employ layerwise normalization. This is, we set the diagonal entries of $T_{w_i} = ||\mathbf{W}_{\text{layer}[i]}||_2$, which corresponds to a normalization with respect to the $l_2$-norm of a layer.

# 3   Method

In this paper, we focus on the interplay between Sharpness-Aware Minimization variants and Batch-Norm. BatchNorm layers transform an input $\mathbf{x}$ with batch mean $\mu_B$ and batch variance $\sigma_B^2$ according to

$$BN(\mathbf{x}) = \gamma \times \frac{\mathbf{x} - \mu_B}{\sigma_B} + \beta$$

During training, $\mu_B$ and $\sigma_B$ are computed from the current batch-statistics, and running estimates are used at test time. In our experiments, we focus on the trainable parameters $\gamma$ and $\beta$, which perform a channel-wise affine transformation. In our first experiment, we exclude them from the adversarial SAM-step (*no-bn*). Taking inspiration from [7], in our main experiment we perturb *only* $\gamma$ and $\beta$ and neglect all other parameters (*only-bn*). *only-bn* corresponds to setting all entries $T_{w_i} = 0$, if $w_i$ is not a BatchNorm parameter, and vice versa for *no-bn*. Apart from this change in $T_w$, which leads to a change of the perturbation $\epsilon$ according to (4), we use the conventional SAM-algorithm.

# 4   Experiments

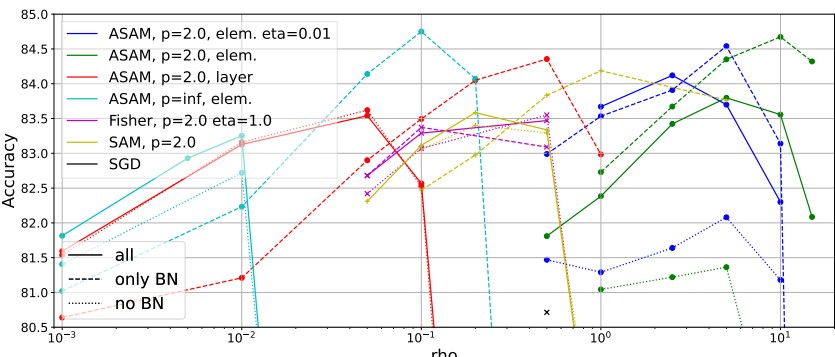

Figure 1: WRN28-10 trained with different SAM-variants on CIFAR-100. Best seen in color.

We first showcase that excluding $\gamma$ and $\beta$ from the $\epsilon$-perturbation (*no-bn*) can lead to drastic performance drops in some variants compared to using all parameters (*all*), but has little effect on others. We train a range of ResNet-like models with several SAM-variants on CIFAR-10 and CIFAR-100 [13] and monitor the performance when computing the perturbation $\epsilon$ only along the directions of $\gamma$ and $\beta$ (*only-bn*). We confirm our findings with experiments on ImageNet [2].

For our CIFAR experiments, we consider a range of SAM-variants which differ either in the norm ($p \in \{2, \infty\}$) or in the definition of the normalization operator. We use SGD, the original SAM with no normalization and $p = 2$, Fisher-SAM and the following ASAM-variants: elementwise-$l_\infty$, layerwise-$l_2$, and elementwise-$l_2$. For each of the ASAM-variants, we normalize both bias and weight parameters and set $\eta = 0$. Additionally, we employ the original ASAM-algorithm, where the bias parameters are not normalized and $\eta = 0.01$. We train all models on a single GPU for 200 epochs, and m-sharpness is not employed. We use both basic augmentations (random cropping and flipping) and strong augmentations (basic+AutoAugment). Like [14], adopt a learning rate of 0.1, momentum of 0.9, weight decay of $0.0005$ and use label smoothing with a factor of 0.1.

We showcase the effect of excluding $\gamma$ and $\beta$ from the adversarial step for a WideResNet-28-10 on CIFAR-100 in Figure 1. For the elementwise-$l_2$ variants we observe a strong drop in accuracy, while for SAM, Fisher-SAM, layerwise-$l_2$ and elementwise-$l_\infty$ there is either no or very little change. If, in contrast, we use *only* the batch-norm parameters for the $\epsilon$ computation, we observe that almost all variants obtain higher accuracy. Except for Fisher-SAM, the difference is especially pronounced for the variants which did not experience a performance drop for *no-bn*. For those, the ideal $\rho$ shifts towards larger values, indicating that the perturbation model cannot perturb the BatchNorm parameters enough when *all* parameters are used. In order to confirm this observation, we train a range of models on CIFAR-10 and CIFAR-100. For each SAM-variant and dataset, we probe a set of pre-defined $\rho$-values (shown in Table 4) with a ResNet-56 and fix the best-performing $\rho$ for all other

models to compare *only-bn* to *all*. We report mean accuracy and standard deviation over 3 seeds for CIFAR-100 in Table 1. On average, *only-bn* outperforms *all* for all considered SAM-variants and layerwise-$l_2$ works particularly well. On CIFAR-10 (Table 3 in the appendix) the results are similar, but slightly less pronounced due to the very high accuracy of all methods.

For ImageNet, we adopt the timm training script ([18]). We train all models for 100 epochs on 8 2080-Ti GPUs with $m = 64$, leading to an overall batch-size of 512. Apart from $\rho$, all hyperparameters are shared for all models and can be found in the appendix in Table 5. Due to the computational cost of training on large-scale datasets like ImageNet, we can only show preliminary results. In particular, we could not run all models with all methods, but instead selected the most promising *only-bn* variants and compared them against the established methods (SGD, SAM, ASAM elementwise $l_2$). For SAM and ASAM, $\rho$ has been tuned by [6] and [14] and we adopt those values. For the methods involving elementwise $l_\infty$ and layerwise $l_2$ with their respective *all* and *only-bn* variant, we probe two $\rho$ values each and report the result of the better one. The results are shown in Table 2, where we observe that the *only-bn* models outperform their *all* counterparts for elementwise $l_2$ and elementwise $l_\infty$. For layerwise $l_2$, the *all* variant achieves higher accuracy, which might be due to the $\rho$-value of the corresponding *only-bn* variant not being properly tuned. Nevertheless, all *only-bn* variants outperform the previously established methods (SGD, SAM, ASAM). For reference, we also show the values reported for ESAM [4] and GSAM [20], two other SAM-variants we did not include in our study.

Table 1: Accuracy on CIFAR-100. Green indicates best performance across all methods, bold values indicate best performance between *all* and *only-bn* within a SAM-variant.

| | SGD | SAM | | elem. $l_2$ $\eta = 0$ | | elem. $l_2$ $\eta = 0.01$ | |
|---|---|---|---|---|---|---|---|
| | all | all | only bn | all | only bn | all | only bn |
| **D100** | 77.01±0.16 | 79.37±0.70 | **79.92**±0.39 | 78.90±0.20 | **79.83**±0.30 | 79.94±0.36 | **80.14**±0.06 |
| **D100** +AA | 79.72±0.49 | **80.69**±0.05 | 79.46±0.18 | **81.30**±0.25 | 80.89±0.22 | 80.84±0.38 | **81.03**±0.29 |
| **RN56** | 72.82±0.31 | 75.07±0.58 | **75.58**±0.44 | 75.05±0.12 | **76.25**±0.05 | 75.54±0.66 | **76.07**±0.22 |
| **RN56** +AA | 75.26±0.24 | **76.33**±0.33 | 76.02±0.34 | **76.51**±0.06 | 76.04±0.33 | 76.49±0.20 | **76.58**±0.44 |
| **RnxT** | 80.16±0.27 | 81.79±0.36 | **82.18**±0.23 | 81.26±0.24 | **82.30**±0.28 | **82.15**±0.33 | 81.90±0.38 |
| **RnxT** +AA | 80.31±0.29 | 82.33±0.54 | **83.19**±0.20 | 82.00±0.29 | **83.20**±0.15 | 82.78±0.14 | **82.87**±0.27 |
| **WRN** | 80.75±0.22 | 83.37±0.30 | **84.17**±0.28 | 82.38±0.18 | **83.67**±0.28 | **83.67**±0.10 | 83.53±0.22 |
| **WRN** +AA | 83.62±0.15 | 85.27±0.18 | **85.50**±0.12 | 84.80±0.30 | **85.43**±0.33 | 85.25±0.38 | **85.41**±0.08 |

| | elem. $l_\infty$ $\eta = 0$ | | layer $l_2$ $\eta = 0$ | | Fisher $\eta = 1$ | |
|---|---|---|---|---|---|---|
| | all | only bn | all | only bn | all | only bn |
| **D100** | 79.32±0.21 | **79.68**±0.20 | 78.25±0.15 | **79.77**±0.27 | 79.05±0.53 | **79.37**±0.16 |
| **D100** +AA | 78.35±0.43 | **79.42**±0.42 | 80.46±0.30 | **81.18**±0.21 | **80.86**±0.21 | 80.79±0.33 |
| **RN56** | 75.36±0.12 | **76.10**±0.15 | 74.63±0.09 | **76.03**±0.32 | 75.27±0.04 | **75.37**±0.12 |
| **RN56** +AA | 74.89±0.39 | **76.19**±0.35 | 76.23±0.54 | **76.93**±0.43 | 76.22±0.29 | **76.29**±0.08 |
| **RnxT** | 81.02±0.59 | **82.39**±0.34 | 81.66±0.22 | **82.46**±0.14 | 81.53±0.14 | **82.03**±0.37 |
| **RnxT** +AA | 82.33±0.12 | **83.11**±0.19 | 82.61±0.31 | **83.32**±0.24 | 82.49±0.17 | **82.74**±0.30 |
| **WRN** | 83.33±0.17 | **84.11**±0.26 | 83.23±0.16 | **84.05**±0.23 | 83.29±0.11 | **83.37**±0.04 |
| **WRN** +AA | 85.28±0.11 | **85.46**±0.13 | 85.40±0.30 | **85.98**±0.01 | **85.13**±0.26 | 84.92±0.31 |

Table 2: ImageNet top-1 accuracy. ESAM and GSAM values are taken from the respective papers.

| top-1 | SGD all | SAM all | ESAM[4] all | GSAM[20] all | elem. $l_2$ all | only bn | elem. $l_\infty$ all | only bn | layer $l_2$ all | only bn |
|---|---|---|---|---|---|---|---|---|---|---|
| Resnet-50 | 77.09 | 77.67 | 77.05 | 77.20 | 77.62 | 77.76 | 77.50 | 77.81 | 78.19 | 77.91 |

## 5  Conclusions and Future Work

In this paper we showed that excluding the BN-parameters in ASAM degrades its performance, whereas performing the adversarial SAM-step *only* for $\gamma$ and $\beta$ brings improvements for a variety of SAM-variants. This provides a practical training tweak, but also casts doubt on the commonly accepted explanation of SAMs empirical success. Since the BatchNorm parameters typically account only for a tiny fraction of the models parameters (for a WideResNet-28-10, $0.05\%$ of all parameters belong to BatchNorm layers), it is unclear if the sharpness quantity used by SAM-variants could still be optimized well with *only-bn*. In addition, the good performance of layerwise normalization, which is not invariant to parameter rescaling, questions the relevance of designing reparametrization-invariant sharpness measures. While we do not have a conclusive answer on why *only-bn* works, we

provide an experiment in Appendix A.3 giving additional insights into the method's impact on BN parameters.

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

# A Appendix

The appendix is structured as follows: In A.1 we show the results of the CIFAR-10 experiment and the $\rho$ values that were used. In A.2 we show the hyperparameters used for the ImageNet runs. In A.3 we show an additional experiment, aimed at providing insight into the effects of *only-bn* on the scale of the BatchNorm parameters.

## A.1 CIFAR-Experiments

Table 3 show the results for fixed $\rho$ on CIFAR-10. The values of $\rho$ we considered for each method can be found in Table 4. The networks we considered for the CIFAR-experiments are DenseNet100 (D100), ResNet56 (RN56), ResNeXt-29-32x4d (RnxT), and WideResNet-28-10 (WRN).

Table 3: Accuracy on CIFAR-10

| | | SGD | SAM | | elem. $l_2$ | $\eta = 0$ | elem. $l_2$ | $\eta = 0.01$ |
|---|---|---|---|---|---|---|---|---|
| | | all | all | only bn | all | only bn | all | only bn |
| **D100** | | $94.51 \pm 0.09$ | $95.76 \pm 0.08$ | $\mathbf{95.88} \pm 0.06$ | $95.76 \pm 0.16$ | $\mathbf{95.86} \pm 0.17$ | $\mathbf{95.92} \pm 0.25$ | $95.85 \pm 0.07$ |
| **D100** | +AA | $95.62 \pm 0.05$ | $\mathbf{96.28} \pm 0.06$ | $96.25 \pm 0.03$ | $\mathbf{96.45} \pm 0.15$ | $96.31 \pm 0.09$ | $\mathbf{96.54} \pm 0.02$ | $96.47 \pm 0.08$ |
| **RN56** | | $94.28 \pm 0.21$ | $94.94 \pm 0.12$ | $\mathbf{95.18} \pm 0.10$ | $\mathbf{94.96} \pm 0.10$ | $94.94 \pm 0.20$ | $95.14 \pm 0.11$ | $\mathbf{95.21} \pm 0.08$ |
| **RN56** | +AA | $94.70 \pm 0.11$ | $95.25 \pm 0.12$ | $\mathbf{95.40} \pm 0.12$ | $\mathbf{95.12} \pm 0.05$ | $94.82 \pm 0.17$ | $95.39 \pm 0.14$ | $\mathbf{95.60} \pm 0.10$ |
| **RnxT** | | $95.37 \pm 0.14$ | $96.35 \pm 0.21$ | $\mathbf{96.48} \pm 0.10$ | $96.41 \pm 0.10$ | $\mathbf{96.53} \pm 0.08$ | $96.41 \pm 0.06$ | $\mathbf{96.41} \pm 0.13$ |
| **RnxT** | +AA | $96.19 \pm 0.22$ | $96.98 \pm 0.11$ | $\mathbf{97.22} \pm 0.27$ | $97.01 \pm 0.05$ | $\mathbf{97.21} \pm 0.12$ | $97.24 \pm 0.04$ | $\mathbf{97.33} \pm 0.11$ |
| **WRN** | | $96.20 \pm 0.05$ | $97.08 \pm 0.08$ | $\mathbf{97.10} \pm 0.04$ | $97.03 \pm 0.23$ | $\mathbf{97.06} \pm 0.04$ | $\mathbf{97.10} \pm 0.08$ | $97.07 \pm 0.13$ |
| **WRN** | +AA | $97.01 \pm 0.04$ | $97.57 \pm 0.09$ | $\mathbf{97.58} \pm 0.05$ | $97.61 \pm 0.01$ | $\mathbf{97.69} \pm 0.04$ | $97.60 \pm 0.02$ | $97.57 \pm 0.05$ |
| | | | el. l-inf | $\eta = 0$ | layer l2 | $\eta = 0$ | Fisher | $\eta = 1$ |
| | | | all | only bn | all | only bn | all | only bn |
| **D100** | | | $95.56 \pm 0.18$ | $\mathbf{95.91} \pm 0.10$ | $95.48 \pm 0.17$ | $\mathbf{95.82} \pm 0.15$ | $\mathbf{95.81} \pm 0.10$ | $95.80 \pm 0.05$ |
| **D100** | +AA | | $96.20 \pm 0.10$ | $\mathbf{96.38} \pm 0.17$ | $96.33 \pm 0.13$ | $96.28 \pm 0.10$ | $\mathbf{96.25} \pm 0.10$ | $\mathbf{96.25} \pm 0.12$ |
| **RN56** | | | $94.93 \pm 0.08$ | $\mathbf{94.96} \pm 0.04$ | $94.95 \pm 0.17$ | $\mathbf{95.07} \pm 0.06$ | $94.97 \pm 0.04$ | $\mathbf{95.05} \pm 0.07$ |
| **RN56** | +AA | | $95.12 \pm 0.12$ | $\mathbf{95.48} \pm 0.35$ | $\mathbf{95.43} \pm 0.25$ | $95.28 \pm 0.13$ | $\mathbf{95.28} \pm 0.19$ | $95.12 \pm 0.04$ |
| **RnxT** | | | $96.06 \pm 0.22$ | $\mathbf{96.22} \pm 0.07$ | $96.07 \pm 0.30$ | $\mathbf{96.46} \pm 0.06$ | $\mathbf{96.31} \pm 0.02$ | $96.14 \pm 0.04$ |
| **RnxT** | +AA | | $96.70 \pm 0.22$ | $\mathbf{96.91} \pm 0.18$ | $96.80 \pm 0.06$ | $\mathbf{96.88} \pm 0.11$ | $\mathbf{97.07} \pm 0.07$ | $96.97 \pm 0.15$ |
| **WRN** | | | $96.95 \pm 0.16$ | $\mathbf{97.00} \pm 0.11$ | $\mathbf{97.02} \pm 0.02$ | $96.96 \pm 0.13$ | $97.05 \pm 0.13$ | $\mathbf{97.12} \pm 0.08$ |
| **WRN** | +AA | | $97.52 \pm 0.09$ | $\mathbf{97.62} \pm 0.09$ | $\mathbf{97.60} \pm 0.04$ | $97.48 \pm 0.06$ | $97.56 \pm 0.09$ | $\mathbf{97.61} \pm 0.08$ |

|  |  | CIFAR-10 | CIFAR-100 |
|---|---|---|---|
| SAM |  | 0.05, **0.1**, 0.25 | 0.05, **0.1**, 0.5, 1. |
| SAM | only-bn | 0.1, **0.5**, 1 | 0.1, 0.5, **1.**, 5. |
| elementwise $l_2, \eta = 0$ |  | 0.5, 1, **2**, 3, 5 | 0.5, **1**, 2.5, 5., 10. |
| elementwise $l_2, \eta = 0$ | only-bn | 0.5, 1, 2, **3**, 5 | 0.5, 1., **2.5**, 5., 10. |
| elementwise $l_2, \eta = 0.01$ |  | 0.1, **0.5**, 1, 5, 10 | 0.5, **1**, 2.5, 5 |
| elementwise $l_2, \eta = 0.01$ | only-bn | 0.1, **0.5**, 1, 5, 10 | 0.5, **1.**, 2.5, 5 |
| elementwise $l_\infty, \eta = 0$ |  | 0.001, **0.005**, 0.01, 0.05 | 0.001, 0.005, **0.01**, 0.05 |
| elementwise $l_\infty, \eta = 0$ | only-bn | 0.01, **0.025**, 0.05, 0.1 | 0.01, **0.05**, 0.1, 0.5 |
| layerwise $l_2, \eta = 0$ |  | 0.005, 0.01, **0.025**, 0.05, 0.1 | 0.001, **0.01**, 0.05, 0.1 |
| layerwise $l_2, \eta = 0$ | only-bn | 0.05, 0.1, **0.25**, 0.5, 1 | 0.1, **0.2**, 0.5, 1. |
| Fisher, $\eta = 1$ |  | 0.005, **0.01**, 0.025, 0.05, **0.1** | 0.05, 0.1, 0.5 |
| Fisher, $\eta = 1$ | only-bn | 0.05, **0.1**, 0.25, 0.5, 1 | 0.05, **0.1**, 0.5 |

Table 4: Search-space for $\rho$. The values used for the the experiments in 1 and 3 is marked bold.

## A.2   ImageNet Experiments

Table 5 shows the hyperparameters for all variants used for ImageNet training. For SGD, SAM and elementwise-$l_2$ we used the hyperparameters from [6] and [14]. For the elementwise $l_2$ and elementwise-$l_\infty$ we tried 2 $\rho$-values per configuration and report the results of the better one (named $\rho$ (reported) in the table). $\rho$ (discarded) refers to the $\rho$ value we probed, but found to perform worse than the other one.

Table 5: Hyperparameters for training from scratch on Imagenet

| param | SGD | SAM | elem. $l_2$ |  |  | elem. $l_\infty$ |  | layer $l_2$ |  |
|---|---|---|---|---|---|---|---|---|---|
|  | all | all | all | only bn |  | all | only bn | all | only bn |
| train epochs |  |  |  |  | 90 |  |  |  |  |
| warm-up epochs |  |  |  |  | 3 |  |  |  |  |
| cool-down epochs |  |  |  |  | 10 |  |  |  |  |
| batch-size |  |  |  |  | 512 |  |  |  |  |
| augmentation |  |  |  |  | inception-style |  |  |  |  |
| lr |  |  |  |  | 0.2 |  |  |  |  |
| lr decay |  |  |  |  | Cosine |  |  |  |  |
| weight decay |  |  |  |  | 0.0001 |  |  |  |  |
| $\rho$ (reported) |  | 0.05 | 1 | 1 |  | 0.001 | 0.005 | 0.005 | 0.05 |
| $\rho$ (discarded) |  |  |  |  |  | 0.01 | 0.05 | 0.05 | 0.5 |
| Input Resolution |  |  |  |  | $224 \times 224$ |  |  |  |  |

## A.3   Additional Experiment

In order to get a better understanding of the impact of *only-bn* on $\gamma$ and $\beta$, we train a WideResNet-28-10 with different SAM-variants and both *only-bn* and *all*. We show the distribution of $|w_i|$, i.e. the parameter magnitudes, at the end of training for different layer types in figure 2. The $y$-axis is shown on log-scale, since most convolutional parameters are almost zero and would make the effects on the BatchNorm parameters invisible. For elementwise $l_2$ there is no visible change in the distribution of the BatchNorm parameters between *all* and *only-bn*. For elementwise $l_\infty$, layerwise $l_2$ and SAM, however, the magnitude of the BatchNorm parameters shifts clearly towards larger values, especially for the weight parameters. We note that this resembles a pattern we observed when comparing the optimal $\rho$-value for *all* and *only-bn* in figure 1: The optimal $\rho$ of elementwise $l_2$ did not change much, whereas for the other considered methods, it shifted towards larger values for *only-bn*. Additionally and in contrast to the other methods, the elementwise $l_2$ variant showed a strong performance decrease in *no-bn*, indicating that it implicitly focuses on perturbing the BatchNorm layers already. We observe a similar, yet weaker effect when comparing basic to strong augmentations: The BatchNorm parameters shift slightly towards larger values (shown for SGD in figure 2). We note that larger BatchNorm parameters do not necessarily indicate a functionally different network, since there are many reparametrization invariances in ReLU networks, some of which ASAM tries to leverage in its perturbation definition (3). Nevertheless, the scale of the network still has an impact on the training dynamics, since other methods like e.g. weight decay depend on it. While this does not provide a conclusive explanation for the success or the underlying mechanism of *only-bn*-SAM,

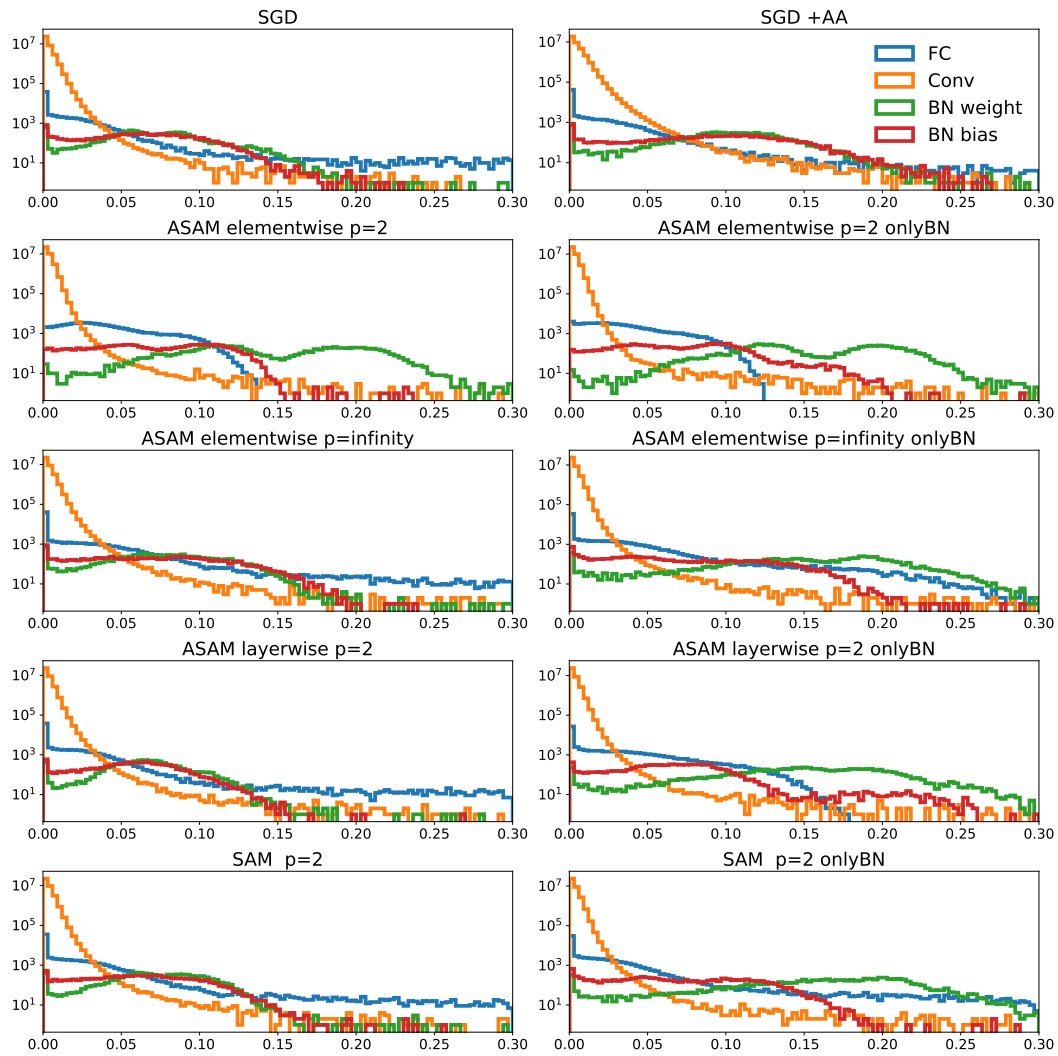

Figure 2: Distribution of $|w_i|$ for different layer types. Note that the $y$-axis is on log-scale. We truncate the $x$-axis at $x = 0.3$ for better visualization.

we think it should be taken as a starting point to investigate the impact of SAM-like methods on other parts of the training of neural networks.

