# OpenReview forum: "Perturbing BatchNorm and Only BatchNorm Benefits Sharpness-Aware Minimization"
_NeurIPS.cc/2022/Workshop/HITY — HITY Workshop NeurIPS 2022_

### Official Review · Reviewer_m8Rb · 2022-10-17
**SAM with batchnorm**

**Rating:** 1
**Confidence:** 4

**Review:**

The paper studies the interplay of SAM with batchnorm, finding that in many situations, the SAM perturbation applied to just the batch norm learnable parameters has increased performance.  This has potential value as a training trick and also is an interesting observation worth investigating more deeply.

---

### Official Review · Reviewer_HGnJ · 2022-10-19

**Rating:** 1
**Confidence:** 3

**Review:**

Overall seems fine to me. Some nits:
- Any sort of model outside the ResNet family would have been nice to see.
- Figure 1 seemed crowded and it would have been useful to exclude some of the less necessary lines. Also the two different styles of dashed lines are hard to pick apart at first. Is it necessary to visualize the trends across \rho, or could you just have used the best \rho value and put them in a table?

---

### Decision · Program_Chairs · 2022-10-20

Accept